# Augmented Reality and Image-Guided Robotic Liver Surgery

**DOI:** 10.3390/cancers13246268

**Published:** 2021-12-14

**Authors:** Fabio Giannone, Emanuele Felli, Zineb Cherkaoui, Pietro Mascagni, Patrick Pessaux

**Affiliations:** 1Department of Visceral and Digestive Surgery, University Hospital of Strasbourg, 1 Place de l’Hôpital, 67100 Strasbourg, France; fabio.giannone@chru-strasbourg.fr (F.G.); e.felli@chu-tours.fr (E.F.); zineb.cherkaoui@chru-strasbourg.fr (Z.C.); 2Institute of Viral and Liver Disease, Inserm U1110, University of Strasbourg, 1 Place de l’Hôpital, 67100 Strasbourg, France; 3University Hospital Institute (IHU), Institute of Image-Guided Surgery, University of Strasbourg, 1 Place de l’Hôpital, 67100 Strasbourg, France; pietro.mascagni@ihu-strasbourg.eu

**Keywords:** robotic, liver surgery, augmented reality, image-guided surgery, artificial intelligence

## Abstract

**Simple Summary:**

Robotic surgery has gained much attention in liver resection for its potential to increase surgical dexterity in a minimally invasive scenario. Different series are reported in the literature with promising results, although strong evidence is lacking. In addition, the robotic system presents the advantage of creating a hybrid interface in which pre- and intra-operative imaging tools could be exploited alone or together in order to guide surgical resection. These technologies have been developed with the aim of increasing surgical safety and improving oncological results. However, some drawbacks are still present, and the literature lacks data given the relatively recent distribution of the robotic platform and of some of these technologies.

**Abstract:**

Artificial intelligence makes surgical resection easier and safer, and, at the same time, can improve oncological results. The robotic system fits perfectly with these more or less diffused technologies, and it seems that this benefit is mutual. In liver surgery, robotic systems help surgeons to localize tumors and improve surgical results with well-defined preoperative planning or increased intraoperative detection. Furthermore, they can balance the absence of tactile feedback and help recognize intrahepatic biliary or vascular structures during parenchymal transection. Some of these systems are well known and are already widely diffused in open and laparoscopic hepatectomies, such as indocyanine green fluorescence or ultrasound-guided resections, whereas other tools, such as Augmented Reality, are far from being standardized because of the high complexity and elevated costs. In this paper, we review all the experiences in the literature on the use of artificial intelligence systems in robotic liver resections, describing all their practical applications and their weaknesses.

## 1. Introduction

Since its presentation, the robotic platform has drawn much attention in liver resections for its possibility to combine the multiple advantages of laparoscopic surgery and the dexterity of an open approach [1,2]. Laparoscopy has widely demonstrated its benefits in terms of shortened recovery and reduced post-operative pain, morbidity, and blood loss, with similar results to open surgery in oncologic outcomes [3,4,5,6,7]. However, some issues are still present when considering technically challenging hepatectomies and most of these disadvantages are related to a limited maneuverability, the presence of rigid instruments, and a restricted field or the quality of vision [8]. Robotic technology is able to overcome some of these limitations and more series and meta-analysis are present in the literature, highlighting the well-known robotic advantages with comparable post-operative and oncological results [9,10,11,12,13,14,15]. Beyond the technical upgrades, one of the improvements of the DaVinci system is the possibility to create an interactive visual interface rather than a simple operative field in which, through some dedicated software, surgeons can be guided by preoperative and/or intraoperative imaging during parenchymal resection [16]. The robot must be considered as a means of putting between the surgeon and the patient a computer and its computing power. Related imaging technologies, such as Augmented Reality (AR), have been developed in order to assist the operator and limit the intrinsic drawbacks of a minimally invasive approach, such as the lack of a tactile feedback, which can hamper tumor detection or pedicle dissection.

In this article, we present an overview of the literature of AR technologies and other strategies of image-guided surgery in the robotic liver scenario. Specifically, we sought to describe all the current clinical and pre-clinical use as well as the future perspectives of imaging technologies used in combination with the robotic platform for treating benign and malign hepatic tumors.

## 2. Augmented Reality

Implementing technologies of robotic hepatic surgery through AR means overcoming the limitations widely described when performing liver resections through this approach. Despite the sense of depth achieved by the 3D visualization and the improved video resolution and magnification, the robotic system presents all the drawbacks of a minimally invasive view compared to an open hepatectomy. The absence of the tactile feedback and the unavailability of an ultrasonic dissector are undoubtedly the main issues found in this context. The importance of a tactile sensation is twofold in this kind of surgery. Firstly, from an oncological point of view, it helps locate the tumor and thus guide parenchymal transection. AR is effective in planning preoperatively the strategy through 3D rendering and, intraoperatively, targeting the lesion and resection margins (Figure 1) [17,18,19,20,21]. Buchs et al. [22], for example, overlaid the 3D preoperative images onto the endoscopic stream and added distance information in order to obtain a real-time guided transection. After projecting the tumor onto the liver surface and defining the margin width, a bar and a dartboard appeared on the screen to guide the tip of the instrument, with colors ranging from green to red to alert the operator if the distance to the tumor was respected or not. Ultrasound guidance or an intraoperative CT scan could be equally used, with the disadvantage of radiation, shifting the instruments constantly, or, occasionally, having difficulty localizing the nodules. In fact, AR has been described in open hepatectomies in the detection of vanishing metastasis, with the superimposition of lesions pre-existent to chemotherapy and undetectable at preoperative images or intraoperative US [23,24]. Secondly, the sense of touch aids the operator to orient him/herself in relation to some intra hepatic landmarks. Arteries, veins, and biliary structures, especially in the first-orders pedicles, present a thickened fibrotic sheath and the robotic “insensibility” can disorient surgeons during the dissection, with possible vascular injuries. Furthermore, lesions can be located in critical areas such as the hepatic confluence, and robotic dissection could thus be more complex [25]. AR-based intraoperative reconstructions and tracking systems may be used to map resection planes and show vascular structures during liver transection (Figure 1). Moreover, 3D planning can enable the identification of anatomical variants, as demonstrated in laparoscopic cholecystectomy [26]. These reconstructions could also represent a solution to the absence of some familiar devices in robotic liver surgery. Recently, a consensus statement suggested that in laparoscopic resections, a deep parenchymal transection must be performed by exposing intraparenchymal structures to avoid blind dissection and major vessel injury [2]. Nevertheless, the ultrasonic dissector and the articulated harmonic scalpel are missing in the robotic ecosystem, and, although no surgical technique has shown its superiority in parenchymal transection [27], some surgeons may be less comfortable without these devices. AR could overcome this gap and act as a support with a non-familiar technique, such as the clamp-crush.

Another advantage of AR described for robotic liver resections is port placement (Figure 2) [28]. This phase is a key step in minimally invasive resections and even more in robotic surgery. Projecting a virtual image of the liver parenchyma on the skin surface in relation to some external landmarks allows targeting the lesion and liver structures at the beginning of the operation for camera placement and, after CO_2_ insufflation, for other ports. This results in an improved manageability of the operator, above all for posterior segment approach, which can lead to fewer operative times and lower intraoperative complications.

### 2.1. External Robot-Assisted Liver Ablation

Radiofrequency and other types of transhepatic ablative procedures are usually proposed in selected cases to treat small hepatic lesions with comparable oncological safety to liver resections [29]. In this scenario, US guidance is normally adopted to localize the lesion and to verify the effect of the treatment. However, US lacks a 3D perception, and during the ablation, the procedure can create a scar or some artefacts that obscure the target, affecting treatment efficiency. Some authors proposed in preclinical studies the utilization of an AR interface to plan the procedure (Figure 3) and guide needle placement through a surgeon–robot cooperative system [30,31]. They concluded that the implementation of a 3D interface combined with a human–computer interaction could improve the safety and accuracy of a percutaneous ablation, although results come from pre-clinical tests.

### 2.2. Augmented Reality in Clinical Practice

Potentialities of these technologies are theoretically limitless and could lead surgery to a simple, automatic, guided gesture. Nevertheless, we are still at the dawn of a new era and the literature lacks important series or prospective comparisons. To our knowledge, only two reports describe the use of AR in robotic liver resections [22,28]. The first report dates back to 2014 and was published by a Swiss group [22]. Two cirrhotic patients underwent an atypical resection for HCC and the authors concluded that the AR interface fit perfectly with the robotic environment and can represent a solution to difficulties in localizing hepatic lesions. Researchers at our center performed with success two robotic segmentectomies V (Figure 1) and one segmentectomy VI for both benign and malignant lesions, finding AR feasible but still under construction and with some drawbacks related to intraoperative registration in a “flexible” model, which limits the superimposition accuracy [28].

## 3. Image-Guided Robotic Liver Surgery

Although AR presents promising results, its application in ordinary life is under development because of the need for standardized algorithms, fully available technologies, and wide clinical data. At present, these drawbacks and the limited distribution of the robotic approach make this fascinating technology less attractive in clinical settings. Other imaging strategies are more frequently used by surgeons during robotic liver resections in order to improve lesion detection and evaluate intraparenchymal biliary and vascular structures. Before exploring these modalities, a special mention goes to TilePro (Intuitive Surgical Inc., Sunnyvale, CA, USA). This is a multi-input display software integrated into the robotic platform, which shows more video sources simultaneously on the same screen (Figure 2). By simply connecting an external source to the Da Vinci console, surgeons and other operating room assistants can easily switch from the operating field to other input such as Intraoperative Ultrasound (IOUS) or preoperative cross-sectional imaging and 3D reconstructions. Furthermore, some authors described the use of this tool during robotic abdominal resections in order to add an intra-operative endoscopic view, e.g., during colonoscopy in low anterior resection, gastroscopy in total gastrectomy, and even common bile duct exploration during left hepatectomy [32].

We next focus on the more common image-based tools and strategies used in combination with the robotic system to perform liver resections.

### 3.1. Preoperative Imaging and 3D Rendering

A correct planification of surgical strategy is essential in liver surgery and it usually includes a 2D cross-sectional preoperative evaluation. However, complex liver intraparenchymal anatomy and anatomical variations can hinder a correct assessment made by a simple CT scan or MRI. Since its first description in 1998 [33], 3D liver reconstruction is nowadays performed more often during preoperative surgical planning. It consists of creating a 3D model starting from preoperative CT scan or MRI, which allows an easier identification of the stereoscopic relationships between the tumor and intrahepatic structures to calculate the hepatic volumes for the risk of post-operative liver failure and create resection plans to follow intraoperatively [34]. These features can be helpful in all types of resections, from anatomical hepatectomies to multiple metastasectomies in the context of a parenchymal sparing approach, or even in extended liver resections. Different retrospective studies found that hepatic surgery based on a preoperative 3D planning had several benefits in terms of operative time and peri-operative complications [18], resection margins [17], vascular resections [35], and oncological outcomes [36] compared to the traditional 2D evaluation. Several software for creating 3D models are on the market or even available as a free-beta, representing a valid solution for simulating a virtual resection and for intraoperative guidance. Despite such potential, liver rendering is still a multi-step and time-consuming technique with a not always optimal quality reconstruction and intraoperative adaptation [37]. Moreover, to our knowledge, no data are present in the literature regarding the use of preoperative 3D images during robotic hepatectomies, and any result in terms of application and perioperative advantages in this field is pure speculation.

### 3.2. Intra-Operative Robotic Ultrasound Application

IOUS is nowadays an essential tool for hepatobiliary surgeons. Several series emphasize its utility in tumor identification, the recognition of its spatial relationship with intrahepatic structures, and, therefore, in guiding parenchymal resection [38]. Furthermore, IOUS is demonstrated to be able to modify preoperative strategy in up to a quarter of cases because of different vascular relationships or new nodules found during surgical exploration, despite the performance of a liver-specific MRI [39,40,41]. Similar results have been described in minimally invasive hepatectomies, with laparoscopic IOUS ensuring adequate perioperative and oncological safety compared to open liver surgery [42,43,44,45].

Originally, the Da Vinci platform did not include a dedicated ultrasound probe. To face this issue, a laparoscopic probe was inserted from the assistant trocar with the double inconvenience of difficult dexterity and the necessity of shifting from the robotic view to the external screen of the ultrasound because of the lack of an integrated system. Recently, a specific transducer was introduced to the robotic ecosystem with consequent better manageability, higher precision, and the creation of a multi-input environment. Surgeons can now easily control the probe, grabbing its dorsal fin with a forceps, and a highly flexible cable and a small transducer surface make the access to the posterior segments or to the inferior surface easier compared to the rigid laparoscopic counterpart [46,47]. Moreover, thanks to software such as TilePro, the operator can shift from the 3D camera view to the ultrasound directly from the console, or even create a split-view with both intraoperative and ultrasound images (Figure 4). Despite all the drawbacks related to the costs and a newborn technology, a few series regarding US-guided robotic liver resection have been published, and results seem promising in terms of surgical and oncological outcomes [48,49].

Other interesting solutions in the field of the US-guider robotic liver surgery have been described in preclinical trials [50,51,52]. A Johns Hopkins group, in order to optimize even more the fusion between US and endoscopic views and thus create an unique multitasking environment, tested an open-source software through which images could be merged in different ways: (i) a split-screen display mode, with an endoscopic US side-by-side view; (ii) a picture-in-picture display mode, in which the US image was inserted in a corner of the camera view; and (iii) a “flashlight” mode, in which a 3D representation of the US image is overlaid in the surgical field on the same plane in which it is physically acquired by the transducer [50]. The two-dimensional (2D) images of the US remain, however, the biggest limitation of this method. Some authors created a 3D US reconstruction by assembling a series of 2D US images and, through a robotic platform, realized an ex vivo microwave liver ablation [51,52]. They concluded that this technique could eliminate error bias, reduce invasiveness (the number of insertions required) compared to manual needle insertion, and provide an accurate estimation of the micro-wave thermal field distribution.

### 3.3. Indocyanine Green Fluorescence

Indocyanine green (ICG) is a fluorescent dye with a rapid hepatic clearance largely used in hepato-biliary surgery thanks to its pharmacokinetics features [53,54]. After portal vein or intravenous injection [55], it allows the identification of anatomical liver vessels and biliary ducts by providing a rapid parenchymal mapping [54]. This enables anatomical resections with lower risks of vascular injuries or bile leaks. Moreover, it increases tumor detectability, helps differentiate hepatic lesions based on their vascular patterns, and allows the detection of additional superficial hepatic lesions [56,57,58,59].

Fluorescence software was incorporated into the Da Vinci system in 2010 and its application enhances robotic advantages in liver surgery from a different point of view. In fact, when a tumor-bearing portal vein injection is chosen—the so-called “negative staining” technique—this could be cumbersome in laparoscopic resections due to the impaired dexterity and lack of ergonomics, and, at the same time, the presence of a rigid linear transducer. The robotic system, through its delicate movements and endowrist instruments, ensures a fine and safe dissection of the hepatic pedicle, allowing it to reach the hilum and the portal bifurcation easier in the case of a direct portal injection [57,58,59,60,61]. Furthermore, through the dedicated probe, the transhepatic needle insertion could be also less demanding.

Another aspect to consider in ICG-guided resections is tumor clearance. Minimally invasive approaches—and robotic, in particular—lack a tactile feedback, and achieving a parenchymal free margin or performing an anatomical resection could be challenging. Furthermore, an IOUS exclusive evaluation could be insufficient because it is a user-dependent procedure and presents a heterogeneous detection rate according to tumor size and location and parenchymal stiffness [62,63,64]. In this context, fluorescence is a precious tool in robotic surgery, with some authors reporting an enlargement of the resection area after ICG application, both in benign ad malignant lesions, in order to achieve a R0 resection [57,65,66,67], and a significantly higher rate of margin-free specimens when comparing robotic hepatectomies with and without ICG [67]. As in open surgery, even in robotic surgery, some series described the detection of newer superficial lesions that the dye injection missed before [57]. This high sensitivity found is, however, limited to the liver surface because of the low penetration of the dye under 8 mm of depth, thus requiring the use of other imaging tools such as IOUS. Although no long-term results have been published, these findings have a significant impact in terms of oncological outcomes.

ICG is a promising instrument of intraoperative navigation surgery, allowing rapid and easy identification of the resection plane without the inconveniences mentioned for other image-guided techniques. It can be used in combination with IOUS or AR as an additional aid rather than as a replacement [61] and with its features, it seems to fill some gaps found in robotic surgery, making tailored and oncological surgery less challenging.

## 4. Future Prospective

The application of the above-described technologies is nowadays limited in the robotic liver experience, mainly due to some technical limitations and to a relatively newborn and still debated approach [15]. AR, for example, is a time-consuming procedure, not only for the intraoperative installation, but also for preoperative planning and liver rendering [68]. In the context of an atypical or less demanding hepatic resection, which represent the first steps of a necessary learning curve, this time could appear exaggerated. Furthermore, AR in hepatic surgery has showed a delayed distribution compared to other surgical fields as neurosurgery, otolaryngology, orthopedics, and maxillofacial surgery [69,70,71]. This difference comes from anatomical obstacles, such as working with a deformable soft organ that is constantly moving during operation because of respiratory cycles as well as pneumoperitoneum creation [72]. Although some strategies have been described in this context [23,73,74], these features make the development of AR more complex, and new software are needed for shortening modeling creation and improving the accuracy of manual, semiautomatic, and automatic images overlapping.

All the imaging techniques described must be seen, however, as a part of a puzzle rather than an independent solution towards a guided surgery; an example comes from registration accuracy in AR. IOUS and ICG have been proposed to improve overlapping quality through fluorescent markers and 3D ultrasounds used for intraoperative landmarks [75,76]. In this scenario, the robotic platform fits perfectly by creating a unique merged environment with the possibility of using and visualizing preoperative reconstruction and intraoperative images simultaneously within the operative field (Figure 4).

Another potential benefit of image-guided technology is minimally invasive training. In laparoscopy, telementoring based on AR seems to speed up simple skills acquisition such as suturing [77] or even reduce the learning curve in more complex procedures such as cholecystectomy [78]. Similar applications in robotic training are lacking, with only a few experiences described [79]. Hepato-biliary surgery lacks standards of training and learning curves in robotic procedures [80], but recently, an expert panel of HPB surgeons agreed that a correct training path in hepatobiliary procedures needs different steps, starting from basic robotic skills before performing a liver resection [81]. In this context, AR could be a useful tool to support less-experienced surgeons performing simple procedures and lower their learning curve.

## 5. Conclusions

The application of pre- and intra-operative imaging modalities in guiding hepatic surgery presents promising results, and the robotic ecosystem can facilitate their use and magnify their benefits. Potential advantages include reduced morbidity and improvements in oncological outcomes. However, some limitations are still present, related to limited robotic diffusion and still insufficient technological development, and most of the data in the literature come from preclinical studies or small series.

## Figures and Tables

**Figure 1 cancers-13-06268-f001:**
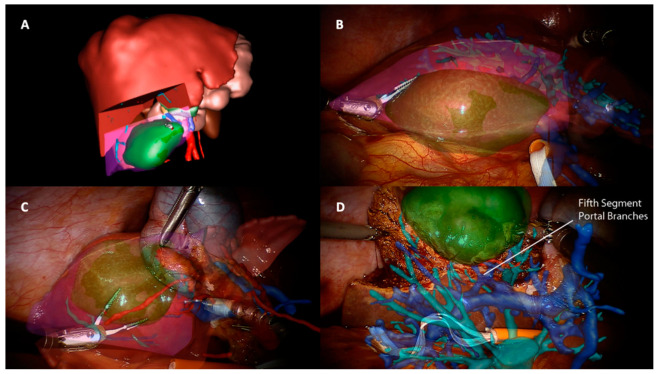
(**A**) Surgical preoperative planning through 3D reconstruction of an anatomical S5 segmentectomy. The tumor is colored in green and the theorical resection plane in red. (**B**–**D**) Intraoperative superimposition of planned resection area rendering. Vascular and biliary structures are projected during different phases of parenchymal transection, with the identification of the S5 vascular pedicle.

**Figure 2 cancers-13-06268-f002:**
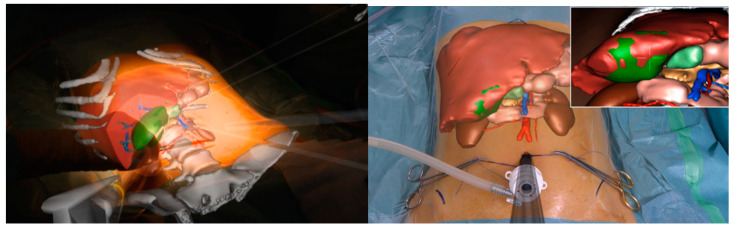
Projection of a virtual liver 3D reconstruction on the skin surface in relation to some external landmarks. The positioning of the optical port (**left**) is guided by the inferior border of the liver and the resection planned. After the first trocar is inserted, the “see-through” view will aid the operator to place other robotic ports (**right**).

**Figure 3 cancers-13-06268-f003:**
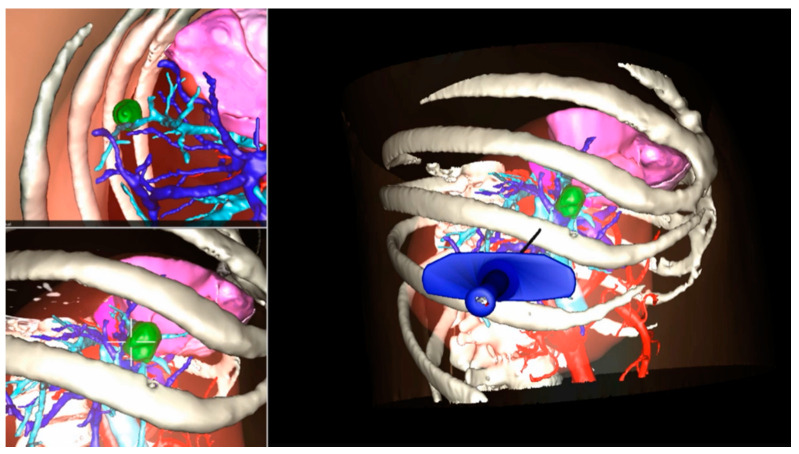
Use of Augmented Reality in planning needle placement during a percutaneous Radiofrequency liver ablation.

**Figure 4 cancers-13-06268-f004:**
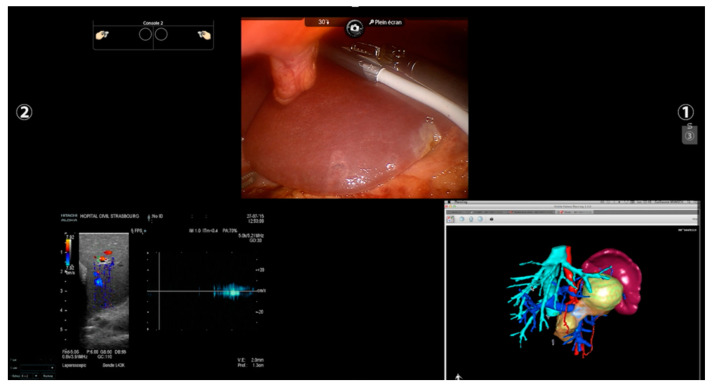
Robotic “split-view”. Through a dedicated probe and a specific software, the surgeon can shift from to the endoscopic to the ultrasound view or create a split-view with both the images. In this figure, a 3D model was added intraoperatively at the same time to check the tumoral vascular relationship studied preoperatively.

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
