# Peer review of "Augmented Reality and Image-Guided Robotic Liver Surgery"

_cancers, 2021, doi:10.3390/cancers13246268_

Round 1
Reviewer 1 Report
In this review the authors reveal actual and potential advantages offered by augmented reality and imaged guided robotic liver surgery. The topics dealt with the complexity of liver surgery, robotic surgery, the possibility to improve some technical challenges thanks to some improvement of technology. All these aspects are rather new and their knowledge presents some interesting food for thought; although this, the contribution offered by the review in terms of deepening the field - actually still unexplored - is only moderate.
To complete the revision, I would suggest to add event the limit of the robotic procedures and a comparison with alternatives strategies.
Author Response
We thank the reviewer for this comment which exactly hits the main limitation of our paper. Unfortunately, our review explores a very niche surgery with several pre-clinical or descriptive studies in literature but a lack of clinical evidences. Robotic surgery in HPB field has started to spread significantly in the last years with always more retrospective series published focusing on perioperative and oncological outcomes. However, Artificial intelligence and image guided resections are still deeply unexplored and the contribution of these technologies, is actually unknown. Costs and time are the main issues in this context which multiply enormously if explored in robotic resections. For example, only two reports are reported, to our knowledge, of hepatic resections with Augmented Reality. At the same time, even if 3D rendering is described widely in other types of hepatic approaches, in the robotic scenario the real contribution of this imaging is lacking.
This study is an invitation review from the journal “Caners” with the aim of exploring the field of image guided surgery in robotic hepatectomies. We though, as the reviewer suggested, to expand the subject by speaking about robotic liver surgery but this would have missed the main aim. We preferred thus to speak of robotic limitations in the context of image guided surgery (i.e. line 63: Although the sense of depth achieved by the 3D visualization and the improved video resolution and magnification, the robotic system presents all the drawbacks…; line 98: Nevertheless, the ultrasonic dissector and the articulated harmonic scalpel are missing in the robotic ecosystem and, although no surgical technique has shown its superiority in parenchymal transection, some surgeons could be less comfortable without these devices…, ecc…).
We agreed that other information on alternative strategies of image-guided surgery should be explored, so we added a paragraph on preoperative imaging and 3D rendering. Similarly, for this technology we found different studies on liver resections and some isolated application in robotic hepatectomies, but no analysis or clinical results. Within the limit of our search, we tried to compare all these strategies by extrapolating conclusions of the various authors.
Reviewer 2 Report
Dear authors,
Thank you for you overview of AI and image-guided surgery options in robotic liver surgery. These techniques willl definitively contribute to patient outcomes in the near future. I do have some concerns about the manuscript. In general the manuscript contains numerous errors in the use of the English language.
Strength is mentioning the TilePro software and describing possibilities using it for multiple (imaging) applications.
The subtopics are being described in little detail, and therefore, the message this manuscript tries to provide the reader remains unclear and vague.
I would suggest to reorder the manuscript, and describe current imaging applications and modalities used in robotic surgery (IOUS, NIR imaging, preoperative CT/MRI) and future perspectives (AR in both surgery and thermal ablation).
Author Response
We thank the reviewer for this comment which exactly hits the main limitation of our paper. Unfortunately, our review explores a very niche surgery with several pre-clinical or descriptive studies in literature but a lack of clinical evidences. Robotic surgery in HPB field has started to spread significantly in the last years with always more retrospective series published focusing on perioperative and oncological outcomes. However, Artificial intelligence and image guided resections are still deeply unexplored and the contribution of these technologies, is actually unknown. Costs and time are the main issues in this context which multiply enormously if explored in robotic resections. For example, only two reports are reported, to our knowledge, of hepatic resections with Augmented Reality (AR).
As the reviewer correctly suggested, we discussed about the TilePro software before exploring all image guided technologies separately. Some data about its use in hepatic surgery is present and added with the appropriate reference. Furthermore, we tried to explore other strategies (paragraph 3.1: preoperative imaging and 3D rendering) and add pros and cons of each technology within the limits of poor clinical evidences.
This study is an invitation from the journal “Caners” with the aim of exploring the field of image guided surgery in robotic hepatectomies. The title as well as the main aim was addressed directly by the editor. We agreed with the reviewer that AR could be consider rather “a future prospective” and described secondarily in short, but the title is “Augmented reality and image guided robotic liver surgery”. For this reason, we focused, with the first paragraph, on the results of AR. We tried to gather all the information available in literature and added other uses, such as thermal ablation (which is not exactly liver surgery) and a separate paragraph of clinical evidences because it is the main subject concerned.
We doubled checked all the text with an english-speaking person and correct all the language mistakes found.
Reviewer 3 Report
Well structured and good overview of the latest data and facts in image guided Robotic Liver Surgery.
The review can address a international audience, pinpointing the main steps and goals in the history of minimal invasive surgery and provides future aspects in imaging techniques.
Author Response
We thank the reviewer for this comment. Although clinical evidences are lacking given the recent spread of robotic liver surgery and an expensive and time-consuming technology, we tried to gather all the data presented in literature about image guided robotic surgery in HPB field. Thank you.
Round 2
Reviewer 1 Report
The authors replied as suggested
Reviewer 2 Report
Dear authors,
Thank you for adding the paragraphs as suggested. The review now provides a clearer story. Please have your newly added paragraphs spell checked one more time.